# Beyond *Trypanosoma cruzi*: LINE-1 Activation as a Driver of Chronic Inflammation in Chagas Disease

**DOI:** 10.3390/ijms26104466

**Published:** 2025-05-08

**Authors:** Marina Dias, Aline Moraes, Tatiana Shiroma, Vitória Pessoa, Antonio Ermoges, Tamires Vital, Luciana Hagström, Davi de Sousa, Márcio de Castro, Bruno Dallago, Izabela Marques Dourado Bastos, Nadjar Nitz, Mariana Hecht

**Affiliations:** 1Interdisciplinary Laboratory of Biosciences, Faculty of Medicine, University of Brasília, Brasília 70910-900, Brazil; marina98dias@gmail.com (M.D.); alinesilvamoraes_df@yahoo.com.br (A.M.); tatiana.ferreira@utsouthwestern.edu (T.S.); vitoriapessoafs@gmail.com (V.P.); antoniovitor25@hotmail.com (A.E.); tamiresvital.biomedicina@gmail.com (T.V.); loubex@hotmail.com (L.H.); dallago@unb.br (B.D.); nadjarnitz@gmail.com (N.N.); 2Veterinary Pathology and Forensics, Faculty of Veterinary Medicine, University of Brasília, Brasília 70910-900, Brazil; daviers@hotmail.com (D.d.S.); mbcastro@unb.br (M.d.C.); 3Laboratory of Pathogen-Host Interaction, University of Brasília, Brasília 70910-900, Brazil; dourado@unb.br

**Keywords:** Chagas disease, *Trypanosoma cruzi*, LINE-1 retroelement, interferons, inflammation, autoimmune response

## Abstract

Chagas disease (CD) is endemic in Latin America, with its pathogenesis linked to *Trypanosoma cruzi* (Tc) persistence and autoimmune responses. This study investigates the role of LINE-1 (L1) activation in inflammation and loss of self-tolerance during Tc infection. In vitro assays evaluated the expression of genes involved in L1 regulation and interferon signaling under basal conditions and following L1 suppression via CRISPR/dCas9. In vivo analyses in a murine model included L1 and IFN expression profiling, autoantibody quantification, and histopathological assessments of liver, spleen, intestine, and heart. Tc infection induced L1 upregulation, correlating with an increased expression of its inhibitors, MOV-10 and APOBEC-3, suggesting host-driven regulatory mechanisms. L1 activation was also associated with the upregulation of DNA repair pathways (MMR and NHEJ) and RNA-sensing pathways (MDA-5 and RIG-I), leading to type I interferon responses. In the murine model, L1 expression was highest in the intestine and heart, independent of parasite burden, and correlated with increased interferon gene expression and autoantibody production. Our findings suggest that CD pathogenesis involves L1-induced chronic inflammation, which may contribute to late-stage symptoms. This highlights self-recognition mechanisms in disease severity and reveals potential therapeutic targets for novel treatments.

## 1. Introduction

Chagas disease (CD), a parasitic infection caused by the protozoan *Trypanosoma cruzi* (Tc), persists as a significant public health challenge in Latin America, where it is endemic across 21 countries. Current epidemiological data suggest an estimated six to seven million individuals are seropositive for Tc, resulting in approximately 12,000 annual fatalities [1]. Following initial infection, the disease typically progresses through an acute phase characterized by elevated parasitemia, although this phase frequently presents asymptomatically [2]. Subsequently, a chronic phase ensues, wherein an estimated 30–40% of infected individuals develop severe clinical manifestations, often decades post-infection. During this chronic stage, despite a substantial decrease in parasite burden, patients may develop the so-called “megasyndromes”, which primarily involve the cardiovascular system and, to a lesser extent, the gastrointestinal tract, often resulting in severe organ dysfunction [2,3,4,5].

The discrepancy between parasite burden and disease severity has led to the hypothesis that Tc persistence alone does not fully account for CD pathogenesis. Instead, autoimmune mechanisms, triggered by molecular mimicry, bystander activation, or even genetic exchanges between the parasite and host, are proposed as critical contributors to disease progression [5,6,7,8]. These processes are believed to induce a breakdown in immunological tolerance, leading to impaired self-recognition and the initiation of pro-inflammatory cascades that cause progressive tissue damage. The resulting chronic inflammatory state ultimately contributes to the functional impairment of affected organs [9,10].

Among the various factors implicated in autoimmune responses, retroelements play a crucial role. These are mobile DNA sequences capable of transposing within the genome, duplicating during the process [11,12]. Notably, the sequencing of the human genome has revealed that transposable elements constitute approximately 45% of the total genetic material, with the LINE-1 (L1) retroelement accounting for 17% of the genome [13]. L1 elements, approximately 6 kb in length, encode three open reading frames (ORFs): ORF0, ORF1, and ORF2. ORF2 encodes the ORF2p protein, which possesses endonuclease and reverse transcriptase activity, while ORF1 encodes ORF1p, a protein with nucleic acid chaperone function [14,15,16]. Although the role of ORF0 remains unclear, it is postulated to regulate L1 mobility [17,18].

L1 activation has been shown to induce the release of significant amounts of cytosolic DNA and RNA, which serve as ligands for innate immune sensors such as cGAS-STING, RIG-I, and MDA-5 [19]. These pathways, in turn, trigger robust interferon type I (IFN-I) responses, further amplifying inflammation [20,21]. Persistent activation of these pro-inflammatory mechanisms has been implicated in the pathogenesis of autoimmune diseases such as systemic lupus erythematosus (SLE) and Sjögren’s syndrome (SS), primarily due to the dysregulation of the Th1/Th2 immune balance [21,22,23]. Consequently, tight regulation of L1 expression is essential to maintaining cellular and systemic homeostasis.

To mitigate the potential deleterious effects of L1 activity, multiple regulatory mechanisms are in place to inhibit retrotransposition at various stages. These include DNA methylation, which limits L1 transcription [24,25,26], RNA degradation pathways that target newly synthesized L1 transcripts [27,28], the translational repression of ORF1 and ORF2 [29,30,31], and DNA repair mechanisms that restrict L1 integration into the genome [29,32,33]. Importantly, DNA repair pathways exhibit a dual role in L1 regulation, as L1 elements can exploit these mechanisms to facilitate their own integration, also contributing to the repair of damaged genomic regions through retrotransposon-mediated repair [32,34,35].

Given the potential involvement of L1 retroelements in triggering aberrant immune responses [18], this study aimed to comprehensively evaluate L1 activation in the context of Tc infection, utilizing both in vitro and in vivo models. Specifically, we investigated the temporal dynamics of L1 expression following Tc infection and its potential role in modulating IFN signaling pathways and contributing to the development of autoimmune manifestations associated with CD. By elucidating these mechanisms, this study provides insights into how L1 activation may influence host–pathogen interactions and immune dysregulation during Tc infection.

## 2. Results

### 2.1. Trypanosoma cruzi Upregulates LINE-1 Expression and Triggers Interferon Production In Vitro

To evaluate the impact of Tc infection on L1 expression, we conducted quantitative qPCR testing on HEK-293 cells from infected and uninfected groups at multiple time points. A significant upregulation of L1 expression was detected at 48 and 96 hpi. A statistical analysis confirmed a highly significant difference between infected and control groups at these time points (*p* < 0.0001; Figure 1A), with a 37.6-fold increase at 48 hpi and a 136.7-fold increase at 96 hpi. These findings indicate that Tc infection induces L1 upregulation, as no comparable alterations were observed in the uninfected control group.

We next examined the temporal expression dynamics of genes associated with L1 regulation (Figure 1B), focusing initially on those implicated in L1 inhibition. MOV-10 exhibited significant upregulation, with relative increases of 3.4- and 7-fold at 48 and 96 hpi, respectively, compared to the control group (*p* < 0.0001). Similarly, APOBEC-3 expression was significantly elevated at 96 hpi (*p* < 0.0001), displaying a 3.8-fold increase. However, at earlier time points, APOBEC-3 was downregulated at 12 and 24 hpi, returning to baseline levels by 48 hpi. In contrast, despite their established roles in restricting L1 mobilization, SAMHD-1 and TREX-1 expression remained unaltered throughout the experimental timeframe.

Among the genes associated with DNA repair pathways (refer to Appendix A for a detailed overview of the analyzed genes and their respective pathways), OGG-1 exhibited no significant changes in expression between infected and control groups throughout the experimental period. Conversely, significant upregulation was observed for ATM (12 and 96 hpi), MSH2 (48 and 96 hpi), XRCC-4 (48 and 96 hpi), and XPA (12 and 24 hpi) in infected cells relative to controls. Specifically, ATM exhibited a two-fold increase at 12 hpi and a 2.7-fold increase at 96 hpi (*p* = 0.0001). MSH-2 expression increased by 1.6-fold at 48 hpi and 1.8-fold at 96 hpi (*p* < 0.0001). XRCC-4 showed a 1.7-fold upregulation at both 48 and 96 hpi (*p* = 0.0195). Notably, XPA displayed the most pronounced induction, with 5.3- and 25.5-fold increases at 12 and 24 hpi, respectively (*p* < 0.0001).

Regarding immune response genes involved in interferon activation, MDA-5, an RNA sensor that triggers type I interferon signaling, was significantly upregulated from 24 hpi onward in infected cells compared to controls (*p* < 0.0001), with relative increases of 2.5-, 19-, and 2.5-fold at 24, 48, and 96 hpi, respectively. Of note, RIG-I expression exhibited a significant upregulation only at 96 hpi, showing a 1.9-fold increase (*p* < 0.0001).

An analysis of interferon gene expression revealed that Tc infection induced a significant upregulation of IFN-α, IFN-β, and IFN-γ over time compared to the control group. IFN-α expression was significantly higher at later time points, with 2.3- and 2.1-fold increases at 48 and 96 hpi, respectively (*p* < 0.001). IFN-β expression increased at 12 hpi (a 4.3-fold increase), returned to baseline at 24 hpi, and exhibited a subsequent surge at later time points, with 11.3- and 4.7-fold increases at 48 and 96 hpi, respectively (*p* < 0.0001). IFN-γ expression was most pronounced at 48 hpi, showing an 8.1-fold increase compared to controls, followed by a six-fold increase at 96 hpi. It is worth noting that, at earlier points (12 and 24 hpi), IFN-γ expression was downregulated in infected cells relative to controls (*p* < 0.0001).

### 2.2. CRISPR/dCas9-Mediated LINE-1 Suppression Attenuates Interferon Response During Trypanosoma cruzi Infection

To determine whether Tc-induced L1 activation is essential for sustaining the IFN-mediated immune response, we utilized transfected cells in which L1 expression was suppressed using the CRISPR/dCas9 system. As shown in Figure 2A, cells transfected with a guide RNA targeting the ORF2 region of L1 (gLINE-1) exhibited distinct bright nuclear foci, whereas negative control cells transfected with gGAL4 displayed only diffuse nuclear GFP fluorescence, confirming the specificity of the targeting approach. A subsequent analysis of L1 expression in HEK-293 cells at 48 hpi revealed a 5.5-fold reduction in L1 transcript levels in gLINE-1-transfected and infected cells compared to the infected-only group (*p* = 0.027), confirming effective gene suppression (Figure 2B).

To assess upstream and downstream effects of L1 inhibition, we analyzed the expression of genes associated with L1 suppression (MOV-10), immune response activation (MDA-5), and interferon production (IFN-α, IFN-β, and IFN-γ). A trend toward reduced MOV-10 expression was observed in the transfected and infected group compared to the Tc-infected group (*p* = 0.09; Figure 2C). Notably, MDA-5 expression was significantly reduced—by 10-fold—in the transfected group relative to the infected group (*p* = 0.039), reaching levels comparable to those of the control group (Figure 2D).

Regarding type I IFNs, IFN-α exhibited a decreasing trend in the transfected and infected group compared to the infected group (*p* = 0.06), while IFN-β expression was significantly downregulated in the transfected group (*p* = 0.05; Figure 2E and Figure 2F, respectively). Similarly, IFN-γ expression was significantly lower in the transfected group, exhibiting a 5.4-fold reduction compared to the infected group (*p* = 0.05; Figure 2G).

### 2.3. LINE-1 Drives Autoimmune Responses in Chagas Disease Independently of Parasite Burden

To evaluate the impact of L1 activation on the pathogenesis of CD, we examined its expression in various organs of BALB/c mice during the chronic phase of Tc infection. First, the parasite load was assessed in the tissues of infected animals (Figure 3A). Statistical analysis revealed significant variation in the Tc load between the heart and intestine (*p* = 0.05), while no Tc DNA was detected in the tissues of uninfected controls.

Subsequently, L1 expression was quantified in the same tissues of infected and control mice (Figure 3B). A statistically significant increase in L1 expression was observed in all infected tissues compared to controls (*p* ≤ 0.05). Specifically, L1 expression was elevated by 3.3-fold in the liver (*p* = 0.03), 79.2-fold in the spleen (*p* = 0.01), 269.4-fold in the intestine (*p* = 0.04), and 682.3-fold in the heart (*p* = 0.04). These findings corroborate the in vitro data, confirming that Tc infection leads to enhanced L1 expression in vivo.

Additionally, our analysis revealed notable differences in L1 expression across the organs of infected animals. The heart and intestine exhibited the highest L1 expression levels, which were statistically equivalent to each other. Specifically, the heart expression was 108-fold higher than the liver (*p* = 0.02) and 7.1-fold higher than the spleen (*p* = 0.02). In the intestine, the L1 expression was 2.4-fold higher than the liver (*p* = 0.04) and 36.4-fold higher than the spleen (*p* = 0.05). The liver displayed the lowest L1 expression, with the spleen showing a 15.3-fold higher expression than the liver (*p* = 0.03).

Next, the in vivo expression of genes associated with the production of distinct interferon subtypes was analyzed. Statistically significant differences (*p* ≤ 0.05) were observed exclusively in the intestine and heart when comparing Tc-infected mice to uninfected controls, whereas the liver and spleen exhibited expression levels comparable to those of the control group.

Regarding IFN-α (Figure 4A), expression in the intestine was 2.5-fold higher in the infected group relative to the control (*p* = 0.02), while in the heart, expression was increased by 6.7-fold (*p* = 0.02). For IFN-β (Figure 4B), expression levels were 4-fold higher in the intestine (*p* = 0.04) and 19-fold higher in the heart (*p* = 0.02) in the infected group. Similarly, IFN-γ expression (Figure 4C) was upregulated by 4.9-fold in the intestine (*p* = 0.01) and by 27.4-fold in the heart (*p* = 0.02) in infected mice compared to controls.

ELISA assays were conducted to assess the adaptive immune response by detecting specific antibodies against Tc and autoantigens from hepatic, splenic, intestinal, and cardiac tissues (Figure 5). All infected animals exhibited IgG production against Tc, whereas in the control group, one animal displayed antibody levels within the indeterminate range, while the remaining animals were seronegative.

Regarding autoantibody production, no animal tested positive for splenic autoantibodies. Similarly, hepatic autoantibodies were undetectable in the infected group, with five animals showing no production and two exhibiting indeterminate results. In contrast, intestinal autoantibodies were detected in all infected animals, whereas none were observed in the control group. Cardiac autoantibodies (i.e., anti-heart IgG) were present in 71% of infected animals (5/7), while all control animals remained seronegative.

To further assess the progression of autoimmune responses, we analyzed the presence of inflammatory infiltrates in the liver, spleen, intestine, and heart of control and Tc-infected mice. Figure 6a presents representative histopathological images of inflammatory infiltration in the intestine and heart.

In the heart, three control animals exhibited mild inflammation, whereas in the infected group, one animal showed mild-to-moderate inflammation, and two displayed moderate inflammation. In the intestine, inflammatory infiltrates were observed exclusively in infected animals, with four cases classified as mild. No inflammatory infiltrates were detected in the spleen. In the liver, one infected animal exhibited moderate inflammation, while one control animal presented mild inflammation. The remaining animals showed no detectable inflammatory infiltrates in this organ (Figure 6b).

Finally, the relationship between the quantitative parameters analyzed and the upregulation of L1 expression in response to Tc infection was assessed through a correlation analysis (Figure 7A). The results indicated no significant correlation between parasite load and the expression of L1, interferons, or the production of autoantibodies in the analyzed organs. However, strong positive correlations were observed among several immune and molecular parameters. Notably, LINE-1 expression exhibited strong positive correlations with IFN-α (r = 0.97), IFN-β (r = 0.90), IFN-γ (r = 0.93), and autoantibody production (r = 0.78).

To further support these findings, a principal component analysis (PCA) was conducted (Figure 7B). Principal components 1 and 2 accounted for 94.9% of the total data variance, with component 1 alone explaining 88.4% of the variation. Importantly, the parasite load was positioned in the upper left quadrant, distinctly separated from the other variables. In contrast, autoantibody production, IFN-γ, IFN-β, IFN-α, and L1 expression clustered in the lower right region, indicating a strong interrelationship among these factors.

## 3. Discussion

CD continues to pose a major public health challenge, with its pathogenesis not fully understood. The disease’s severe clinical manifestations emerge decades after the initial Tc infection, despite a low parasite burden, suggesting that factors beyond direct parasite damage contribute to its progression [2,36,37]. In this study, we showed that L1 retroelement activation shapes pro-inflammatory and autoimmune responses in Tc infection, particularly in affected tissues.

To evaluate the impact of Tc infection on L1 expression in vitro, we quantified L1 mRNA levels at multiple time points. A significant upregulation was observed at 48 and 96 hpi. To the best of our knowledge, this is the first report demonstrating L1 activation in response to a parasitic infection. This phenomenon may be driven by oxidative stress and DNA damage, both of which are well-established triggers of L1 activation [38,39]. Additionally, a hypothetical alternative mechanism could involve the transfer of mitochondrial DNA (kDNA) sequences from the parasite to the host genome, considering that L1 elements have been identified as potential major integration sites for kDNA [40,41].

We also examined genes involved in L1 inhibition, such as MOV-10, a helicase that recognizes L1 RNA and prevents its retrotransposition in an RNA-dependent interplay between MOV-10 and L1, thereby contributing to genomic stability [42,43]. MOV-10 expression increased in line with L1 activity, indicating its role in trying to regulate L1 during Tc infection. Similarly, APOBEC-3, a key inhibitor of L1 retrotransposition through multiple mechanisms [44], was also upregulated at 96 hpi. These findings indicate that APOBEC-3 and MOV-10 are jointly working in concert to mitigate the harmful effects of unchecked L1 retrotransposition, underscoring a coordinated cellular response aimed at preserving genomic integrity during Tc infection.

We also analyzed SAMHD-1, a key regulator of L1 restriction that limits reverse transcription by controlling the availability of dNTPs [44]. SAMHD-1 expression remained unaltered throughout the experimental period, potentially to support DNA repair mechanisms activated in response to infection [45], as an adequate dNTP supply is essential for DNA synthesis. In this context, the precise regulation of SAMHD-1 is likely crucial for maintaining genomic integrity, facilitating cell cycle progression, and ensuring cell survival. Similarly, TREX-1 expression remained unchanged over time, suggesting a minimal role in modulating L1 activity during Tc infection.

To further explore the regulatory mechanisms influencing L1 activity, we analyzed the expression patterns of genes involved in DNA repair pathways, which play a dual role in L1 modulation, either facilitating retroelement insertion or restricting its mobility to maintain genomic integrity [32,35,46]. A temporal analysis of our data revealed that, during the early stages of infection, there was an increase in XPA and ATM expression, indicating the activation of the nucleotide excision repair (NER) and homologous recombination (HR) pathways, respectively.

Although primarily activated to mitigate the effects of parasitic infection and DNA damage, these pathways may also regulate L1 activity. The NER pathway is known to suppress L1 retrotransposition by facilitating the interaction of the ERCC1-XPF protein complex with elongating L1 cDNA, leading to its cleavage [46]. Similarly, components of the HR pathway can inhibit retrotransposition by promoting BRCA1 binding to L1 RNA in the cytoplasm, thereby reducing L1-ORF2p synthesis [32]. Thus, these mechanisms may provide insights into the lack of L1 activation observed until 24 hpi. Additionally, the upregulation of ATM at 96 hpi may suggest that, at this later stage, the activation of the HR pathway may contribute to restraining L1 activity.

XRCC-4, a key component of the non-homologous end joining (NHEJ) pathway, exhibited sustained upregulation precisely during the period of highest L1 expression. L1 is known to exploit the NHEJ machinery to facilitate its genomic integration [34], and the inhibition of this pathway has been shown to reduce L1 activity [47]. Similarly, the mismatch repair (MMR) pathway, assessed through MSH-2 expression, was also upregulated at 48 and 96 hpi, suggesting a potential role in modulating L1 activity. However, the direct involvement of the MMR pathway in L1 regulation has not been characterized in the literature.

Finally, we evaluated the expression of OGG-1, which remained unaltered throughout the experimental period. The base excision repair (BER) pathway is typically activated during the early stages of Tc infection [38,48]; thus, its involvement may not have been detected due to the time points analyzed, which began at 12 hpi. The absence of significant changes in OGG-1 expression further suggests that this repair mechanism may not play a major role in modulating L1 activity under the experimental conditions tested.

Given that the L1 retroelement modulates the expression of interferon-stimulated genes [19,49], we next examined the transcriptional dynamics of these genes in the context of Tc infection. The RNA sensors MDA-5 and RIG-I, which play critical roles in the recognition of double-stranded RNA (dsRNA), exhibited differential expression patterns throughout the course of infection. This finding is consistent with previous studies that have reported the upregulation of different RNA sensors during Tc infection [50,51,52]. The periodic activation of MDA-5 (at 12 h, 48 h, and 96 h) suggests a potential association with distinct stages of L1 retrotransposition in host cells [33,53]. Since MDA-5 primarily recognizes long dsRNA structures [54], it is plausible that this sensor detects dsRNA intermediates generated during L1 activation [55]. This hypothesis is further supported by the observed reduction in MDA-5 expression following L1 suppression via the CRISPR/dCas9 system, reinforcing the link between L1 activity and innate immune activation.

In contrast, RIG-I, which preferentially detects short dsRNA molecules [56], was upregulated only at 96 hpi. This delayed response may reflect the recognition of L1-derived small RNA fragments, potentially generated through processing by DNA repair pathways, or may result from the activation of smaller non-autonomous retroelements, such as Alu elements [56,57]. These findings suggest a dynamic interplay between L1 activity and the host antiviral response, with distinct RNA sensors being engaged at different stages of infection and retrotransposition.

A marked increase in the production of the evaluated interferons was observed in the infected group, with expression peaks consistently occurring after 48 hpi, coinciding with the upregulation of L1 expression. Regarding type I interferons, IFN-β exhibited higher expression levels, aligning with previous reports in the literature that describe its predominant induction in retrotransposition contexts [58]. In contrast, IFN-α expression was relatively lower, likely due to its dependence on the positive feedback loop triggered by IFN-β [59].

IFN-γ plays a crucial role in the immune response against Tc, acting as a key cytokine in infection control [60]. However, its production in HEK-293 cells is typically negligible, as these cells are deficient in the JAK/STAT signaling pathway [61,62]. Despite this inherent limitation, we observed a significant increase in IFN-γ expression at 48 hpi, with L1 involvement confirmed by CRISPR/dCas9 experiments. These findings suggest that IFN-γ activation during Tc infection may occur independently of canonical JAK/STAT signaling, potentially engaging an alternative cytosolic surveillance mechanism mediated by L1. If this hypothesis is confirmed, it may support a parasite-independent pro-inflammatory response, reducing the need for sustained Tc presence to maintain immune activation. Further investigations are required to elucidate the molecular mechanisms underlying this phenomenon and to determine whether L1-driven IFN-γ expression contributes to broader antiparasitic defense strategies.

Extending our investigation to the in vivo model, a gene expression analysis of murine tissues revealed a significant upregulation of L1 mRNA in infected mice compared to controls. The highest activation was detected in the heart, followed by the intestine—both primary sites of pathology in CD [2,4] and tissues with inherently elevated basal L1 expression [63]. Notably, L1 upregulation appeared to occur independently of the local parasite burden, as indicated by comparable Tc loads across the analyzed organs. This finding was further supported by correlation and principal component analyses, suggesting that L1 activation is not merely a secondary effect of parasite presence but may play an active role in shaping the pathophysiological landscape of CD.

Given that retroelements’ activation can amplify immune responses, contributing to chronic inflammation and potentially increasing susceptibility to autoimmune diseases [20,21,64], we next assessed the expression of IFNs in the same tissues previously analyzed. Our findings revealed that the upregulation of these proinflammatory genes was restricted to the heart and intestine. In contrast, no statistically significant differences were observed in spleen and liver expression levels when compared to the control group. These results reinforce the hypothesis that L1 activation and its immunomodulatory effects are tissue-specific, occurring preferentially in sites of chronic inflammation rather than systemically. This localized inflammatory response may contribute to the pathogenesis of CD, highlighting a potential role for L1-induced immune signaling in disease progression.

The absence of IFN induction in the liver and spleen may be attributed to the predominance of an endogenous L1 variant generated through alternative splicing, which undergoes premature polyadenylation, rendering it functionally incomplete [63]. Consequently, the downstream pathways typically activated by the retroelement may be compromised, limiting its capacity to trigger an immune response. Conversely, in the heart and intestine, the expression of a full-length, transcriptionally active L1 variant is predominant, potentially facilitating its immunomodulatory effects and contributing to the localized inflammatory response observed in these tissues [63]. Therefore, RNA processing can be regarded as a key regulatory mechanism of L1 activity across different tissues.

The persistence of an inflammatory microenvironment driven by IFN expression has been implicated in the initiation and progression of autoimmune responses, characterized by inflammatory infiltrates and the production of autoreactive antibodies [65,66]. Notably, type I IFNs play a pivotal role in B cell activation, promoting the synthesis of autoantibodies [67,68,69]. In particular, IFN-α has been extensively associated with autoimmune pathogenesis, as observed in systemic lupus erythematosus (SLE), in which it enhances B cell differentiation and the production of autoantibodies targeting DNA and nucleosomes [69]. Additionally, IFN-β has been implicated in autoimmune disorders such as multiple sclerosis, in which its dysregulation contributes to immune-mediated neuroinflammation [70].

In this study, ELISA assays confirmed the presence of autoantibodies exclusively targeting cardiac and intestinal tissues, aligning with previous findings [71,72]. Remarkably, correlation analyses revealed a strong association between autoantibody production, L1 expression, and IFN signaling, suggesting a potential mechanistic link between L1 activation and the autoimmune responses observed in CD. These findings reinforce the hypothesis that L1-induced IFN production may contribute to the breakdown of immune tolerance, promoting the generation of autoreactive antibodies and the establishment of a chronic inflammatory state in affected tissues.

Regarding the inflammatory response, the most significant findings were observed in the intestinal and cardiac tissues of infected animals. However, these inflammatory alterations were not as pronounced, which may be attributed to the intrinsic limitations of the murine model in accurately replicating the severity of clinical manifestations observed in human CD [73]. Additionally, the animals were still in the early stages of the chronic phase, a period in which overt clinical manifestations are not yet fully established [74,75]. Despite this, it is important to highlight that previous studies have reported extensive cardiac lesions in mice infected with the Colombiana strain [72,76]. Furthermore, the inflammatory process was detected specifically in the organs that exhibited a marked increase in L1 expression and IFN production, occurring independently of the parasitic load. These findings further support the hypothesis that the L1 retroelement plays a role in modulating chronic inflammation and the autoimmune response in CD.

## 4. Materials and Methods

### 4.1. Experimental Groups

This study compared Tc-infected and uninfected control groups using both in vitro and in vivo models. Three experimental conditions were established for the in vitro assays: (i) uninfected control cells, (ii) Tc-infected cells, and (iii) L1-inhibited cells transfected prior to infection. All groups were cultured in triplicate in DMEM (Dulbecco’s Modified Eagle Medium) supplemented with 2% fetal bovine serum (FBS) and maintained under standard conditions.

For in vivo experiments, a total of 13 female BALB/c mice (4–5 weeks old, weight-matched) were used and assigned to two experimental groups: uninfected negative control (*n* = 6) and Tc-infected (*n* = 7). The animals were housed in ventilated cages (Alesco, Viracopos, SP, Brazil) within the vivarium of the School of Medicine, University of Brasília, under controlled conditions (12 h light/dark cycle and ambient temperature of 22 ± 1 °C) with ad libitum access to food and water. This study was conducted in compliance with the ethical guidelines of the National Council for the Control of Animal Experimentation (CONCEA) and was approved by the Ethics Committee for the Use of Animals at the University of Brasília (CEUA/UnB) under protocols 23106.082680/2017-45 and 122/2019.

### 4.2. Cell and Parasite Culture

In this study, human embryonic kidney cells (HEK-293) and rat myoblast cells (L6) were utilized. HEK-293 cells constituted the primary experimental model, while L6 cells were employed for the propagation of Tc trypomastigote forms. Both cell lines were maintained in DMEM, supplemented with 10% fetal bovine serum (FBS) and 1% penicillin/streptomycin, and adjusted to pH of 7.2. Cultures were incubated at 37 °C in a humidified atmosphere containing 5% CO_2_. The culture medium was replenished every 48 h, and cells were subcultured weekly upon reaching 60–70% confluence using a specific inverted transfer method.

The Colombiana strain of Tc was selected for our study due to its well-characterized myotropic behavior [76]. Epimastigote forms were maintained in Liver infusion tryptose (LIT) medium supplemented with 10% FBS, 100 IU/mL penicillin, and 100 μg/mL streptomycin at 25 °C. Trypomastigote forms were obtained through infection of the L6 cells. Briefly, L6 cells were seeded and cultured in Dulbecco’s Modified Eagle Medium (DMEM) supplemented with 10% (*v*/*v*) heat-inactivated fetal bovine serum (FBS), 100 IU/mL penicillin, and 100 µg/mL streptomycin, under a humidified atmosphere of 5% CO_2_ at 37 °C. Once a confluent monolayer was achieved, the cells were infected with epimastigote forms at a parasite-to-host cell ratio of 5:1. After incubation for ~5 days, trypomastigotes released into the supernatant were harvested, centrifuged at 5000 rpm for 15 min to remove cell debris, and used for subsequent experimental procedures.

### 4.3. Infection with Trypanosoma cruzi

HEK-293 cells were infected with trypomastigote forms of Tc after adhering to T75 culture flasks. Parasite concentration was determined using a Neubauer chamber, and infection was established at a multiplicity of infection (MOI) of 5:1 (parasites per host cell). Cultures were maintained under standard conditions at 37 °C in a humidified atmosphere with 5% CO_2_, as described previously.

For in vivo infection, BALB/c mice were inoculated intraperitoneally with Tc trypomastigotes at a dose of 5 × 10^5^ parasites per animal. The infection was confirmed by monitoring peripheral blood parasitemia from day 7 post-infection (dpi). Blood samples were obtained via tail vein incision and examined under an optical microscope (40× objective). A drop of blood (~10 µL) was mixed with an anticoagulant solution (sodium citrate; Sigma-Aldrich, St. Louis, MO, USA) and placed on a glass slide, followed by coverslip application for microscopic evaluation.

### 4.4. CRISPR/dCas9 System and Cell Transfection

The CRISPR/dCas9 SunTag system utilized in this study consists of three plasmids designed to express a dCas9 enzyme fused to GCN4 peptides, GFP-tagged anti-GCN4 antibodies, and a guide RNA (gRNA) cloning vector. The plasmids employed were pHRdSV40-NLS-dCas9-24xGCN4_v4-NLS-P2A-BFP-dWPRE (Addgene #60910), pHR-scFv-GCN4-sfGFP-GB1-dWPRE (Addgene #60907), and gRNA_Cloning Vector (Addgene #41824), all obtained from Addgene (Cambridge, MA, USA). Plasmid extraction and purification were carried out using the Invitrogen™ PureLink™ HiPure Plasmid Midiprep Kit (Thermo Fisher Scientific^®^, Waltham, MA, USA, catalog K210004), following the Midiprep protocol.

The gRNA_Cloning Vector (Addgene #41824) was linearized using the restriction enzyme AflII, and oligonucleotides were inserted into the vector at a 10:1 molar ratio (oligo:vector) using the NEBuilder™ HiFi DNA Assembly Cloning Kit (New England BioLabs, Ipswich, MA, USA), in accordance with the manufacturer’s guidelines. Double-stranded single guide RNAs (sgRNAs) were designed to specifically target the conserved region encoding the reverse transcriptase domain within ORF-2 region of L1 sequences (sgLINE1: 5′ GAACCAAAAAAGAGCCCACA 3′), as well as a non-targeting control sequence unrelated to the human genome (sgGAL4: 5′ GAACGACTAGTTAGGCGTGTA 3′).

Cells were transfected with plasmids using Lipofectamine 3000 (Thermo Fisher Scientific^®^, Waltham, MA, USA). A day before transfection, 4 × 10^4^ cells were seeded onto 12 mm coverslips in a 24-well plate to allow for adhesion. Each transfection reaction contained 600 ng of total DNA, following a molar ratio of 5:1:1 (gRNA_Cloning Vector #41824: dCas9-GCN4 #60910: scFv-GCN4-GFP #60907). The reagent volumes were maintained at a ratio of 1 µg DNA: 2 µL P3000 reagent: 3 µL Lipofectamine 3000. Quantitative PCR (qPCR) analyses were performed to verify the efficacy of L1 inactivation mediated by the CRISPR/dCas9 system.

At 48 h post-transfection (hpi), the cells were infected and maintained for an additional 48 h before being fixed with 4% paraformaldehyde. Nuclear staining was performed using 300 mM DAPI, and the samples were mounted on microscope slides with ProLong™ Glass Antifade Mountant (Invitrogen™, Waltham, MA, USA). Fluorescence images were captured using an Olympus BX51 microscope with an exposure time of 200 ms.

### 4.5. Sample Collection

For in vitro assays, sample collection was conducted at 12-, 24-, 48-, and 96 hpi, with the exception of transfected groups, for which samples were collected exclusively at 48 hpi.

For in vivo studies, sample collection occurred at the early chronic phase [77,78], at 90 dpi, or at a corresponding time for uninfected animals. Mice were anesthetized with a combination of 10% ketamine hydrochloride and 2% xylazine hydrochloride, administered according to body weight [79]. Blood samples were then obtained via cardiac puncture for serum collection.

Euthanasia was performed using an overdose of the anesthetic mixture (ketamine/xylazine), followed by a ventral incision to extract the liver, spleen, intestine, and heart. A portion of each organ was mechanically dissociated on ice using scalpels and transferred into microcentrifuge tubes containing 1 mL of RNAlater™ solution (Invitrogen™, Waltham, MA, USA), then stored at −80 °C for subsequent RNA extraction. Another fraction of the tissues was fixed in 10% buffered formalin for histopathological analysis.

### 4.6. RNA Extraction and cDNA Synthesis

RNA extraction was performed using the SV Total RNA Isolation System kit (Promega^®^, Madison, WI, USA), according to the manufacturer’s instructions. RNA purity and concentration were assessed using a NanoDrop 2000/2000c spectrophotometer (Thermo Scientific^®^, Waltham, MA, USA), and samples were stored at −80 °C for subsequent analyses. For complementary DNA (cDNA) synthesis, RNA input was standardized to the lowest concentration sample, set at 30 ng/μL, to ensure consistency across all reactions. Reverse transcription was performed using the High-Capacity RNA-to-cDNA™ Kit (Applied Biosystems™, Waltham, MA, USA) according to the manufacturer’s instructions. The reaction was conducted in a T100™ thermocycler (Bio-Rad, Hercules, CA, USA), and the synthesized cDNA was stored at −20 °C until further quantitative PCR (qPCR) analysis to evaluate gene expression.

### 4.7. Quantitative PCR (qPCR)

Gene expression analysis was performed using qPCR on a QuantStudio 3 system (Thermo Scientific^®^, Waltham, MA, USA). Each reaction contained 100 ng of cDNA, 10 μL of GoTaq^®^ qPCR Master Mix 2X (Promega^®^, Madison, WI, USA), and a final reaction volume of 20 μL. The thermal cycling conditions were as follows: initial denaturation at 95 °C for 10 min, followed by 40 amplification cycles of 95 °C for 15 s, target-specific annealing temperature (T °C) for 30 s, and extension at 72 °C for 10 s.

The sequences, concentrations of primers, and specific annealing temperatures used in the study are detailed in Appendix A. All qPCR reactions were performed in technical triplicates. Gene expression was normalized using the 2^−ΔΔCt^ method as described by Livak and Schmittgen (2001) [80].

The parasite burden in organ samples from infected mice was determined by quantifying the Tc40S constitutive gene of Tc (Appendix A). Absolute quantification was performed using a standard curve generated from a ten-fold serial dilution (1:10) of parasite cDNA [38].

### 4.8. Indirect Enzyme-Linked Immunosorbent Assay

The presence of parasite-reactive or autoreactive immunoglobulin G (IgG) was assessed by enzyme-linked immunosorbent assay (ELISA) using serum samples. Briefly, ELISA plates were coated with antigen solutions in 1X PBS buffer (pH 7.4) at a concentration of 15 μg/mL for tissue-derived antigens (liver, spleen, intestine, and heart) or 8 μg/mL for Tc antigens. The plates were incubated overnight at 37 °C in a humidified atmosphere to facilitate antigen adsorption. Following incubation, 150 μL of blocking buffer (1X PBS, pH 7.4, supplemented with 5% *w*/*v* skim milk) was added to each well to minimize non-specific binding. Serum samples were diluted 1:100 in 1X PBS (pH 7.4) containing 2% *w*/*v* skim milk and subsequently applied to the wells. After a 2 h incubation at 37 °C in a humid chamber, plates were thoroughly washed to remove unbound components. Detection was performed by adding 50 μL per well of anti-mouse IgG conjugated to alkaline phosphatase (Sigma-Aldrich) at a 1:1000 dilution. Following a 12 min incubation in the dark, absorbance was measured at 405 nm using a Synergy HT spectrophotometer (BioTeK^®^, Santa Clara, CA, USA). The cut-off values for each antigen were determined based on the mean optical density (OD) of negative controls plus three times the standard deviation (MEAN + 3 × SD [72]). Samples with an OD value exceeding the cut-off by more than 10% were classified as positive, whereas those below the cut-off were considered negative.

### 4.9. Histopathological Analysis

Tissue samples from the liver, spleen, intestine, and heart were fixed in 10% buffered formalin, followed by sequential dehydration in graded ethanol solutions, clearance with xylene, and embedding in paraffin. Serial 5 μm sections were prepared using a Leica RM2125 RTS microtome (Leica Biosystems, Nubloch, Germany). The sections were subsequently stained with hematoxylin and eosin (H&E) and examined under an Olympus CX41 3D optical microscope (TurboSquid, New Orleans, LA, USA) equipped with a 40× objective lens. Image acquisition was performed using the ScanScope system (Aperio^®^, Vista, CA, USA).

Histopathological evaluation included both qualitative and semi-quantitative analyses of tissue responses. The inflammatory process was assessed based on the presence and distribution of mononuclear infiltrates, particularly multifocal perivascular infiltrates and interstitial infiltrates in the organs. The severity of inflammation was scored as follows: 0 (absence), 1 (mild; affecting 2–15% of the total tissue section), 2 (moderate; affecting 16–60% of the total section), and 3 (severe; affecting >60% of the section).

### 4.10. Statistical Analysis

Statistical analyses were performed using SAS^®^ software (v9.4, SAS Institute Inc., Cary, NC, USA), with a significant threshold of 5% (α = 0.05). Data normality was assessed using the Shapiro–Wilk test.

For in vitro gene expression analysis, a repeated-measures ANOVA was conducted to evaluate temporal variations. For post-transfection infected samples, the Kruskal–Wallis test was applied to compare gene expression levels among the control, Tc-infected, and transfected-infected groups, considering a fixed collection time of 48 hpi.

In in vivo experiments, the Kruskal–Wallis test was used to compare gene expression levels across different organs between Tc-infected and control mice. Additionally, the Wilcoxon test was performed for pairwise intragroup comparisons to identify significant expression differences among organs. For the analysis of the humoral immune response, the Shapiro–Wilk test confirmed data normality. Consequently, Student’s *t*-test was applied to compare the production of anti-Tc antibodies and autoantibodies against liver, spleen, intestine, and heart antigens between control and infected groups. To investigate associations among variables, Pearson correlation analysis (PROC CORR) and principal component analysis (PCA) were conducted.

## 5. Conclusions

Our findings reveal a previously unrecognized role for L1 retroelement activation in the pathogenesis of CD. By demonstrating that L1 upregulation occurs independently of the parasite burden and is closely associated with tissue-specific inflammatory responses, we provide evidence that L1-driven immune signaling contributes to chronic disease progression. The activation of interferon-stimulated genes and the production of autoantibodies suggest that L1 retrotransposition may act as a key driver of immune dysregulation, potentially exacerbating tissue damage in affected organs. Moreover, our study highlights the coordinated response of host restriction factors, such as MOV-10 and APOBEC-3, in mitigating L1 activity, underscoring the complex interplay between retroelement regulation and immune homeostasis during Tc infection.

Despite the advances in understanding host L1 activation in response to Tc infection, several critical aspects remain to be elucidated. Future studies should investigate whether L1 activation can also be triggered by different strains, non-viable parasites, or parasite-derived extracellular vesicles. In addition, assessing L1 expression in human tissues is crucial to overcome the inherent limitations of murine models to fully determine the pathophysiology of CD. Notably, our findings open new avenues for therapeutic interventions targeting L1 activity as a means to modulate the inflammatory and autoimmune components of CD. Further investigations are warranted to determine whether pharmacological inhibition of L1 retrotransposition represents a viable strategy to attenuate disease progression. By bridging the fields of retroelement biology and infectious disease pathogenesis, this study contributes to a deeper understanding of host–parasite interactions and their long-term impact on genomic stability and immune regulation.

## Figures and Tables

**Figure 1 ijms-26-04466-f001:**
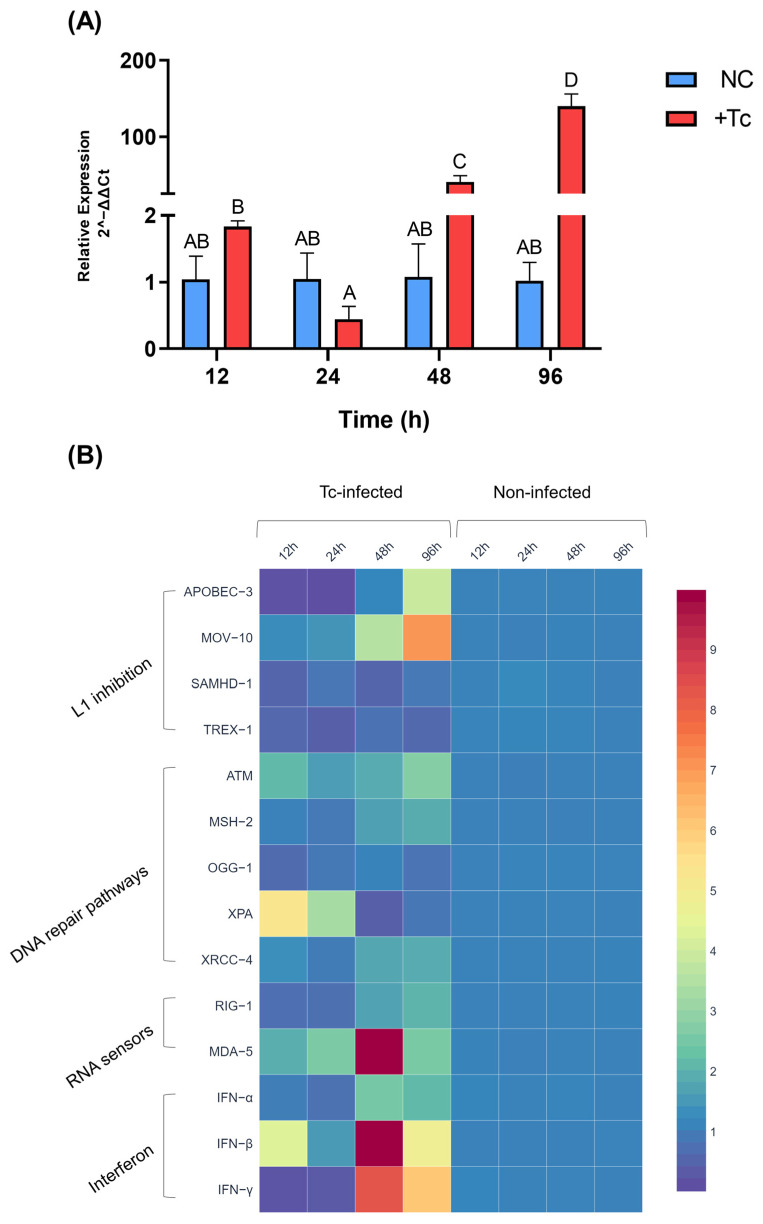
*Trypanosoma cruzi* induces LINE-1 overexpression and activates interferon signaling in vitro. (**A**) Relative expression of LINE-1 in Tc-infected (red) and non-infected (blue) HEK-293 cells over time. Different letters indicate statistically significant differences between groups (*p* < 0.0001). (**B**) Heatmap depicting the relative expression of genes involved in LINE-1 regulation, including inhibitory factors, DNA repair pathways, nucleic acid sensors, and interferons. Gene expression levels were calculated using the 2^−ΔΔCt^ method. NC: negative control, non-infected cells. +Tc: cells infected with *T. cruzi*.

**Figure 2 ijms-26-04466-f002:**
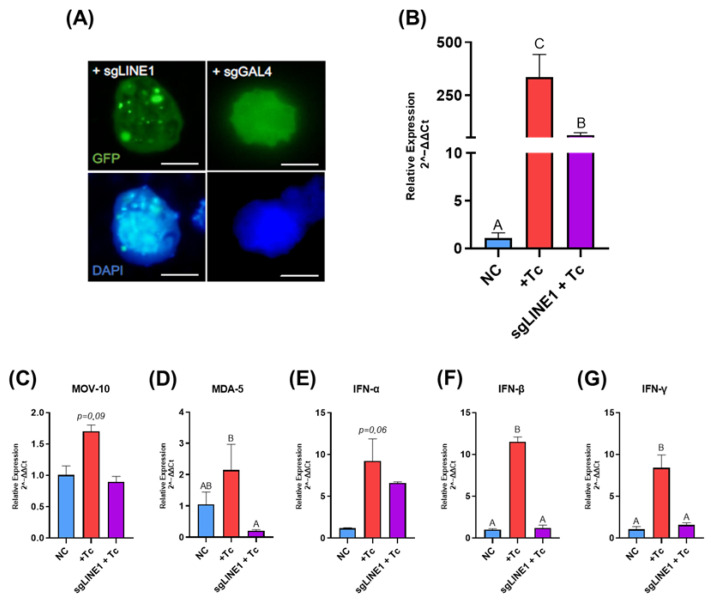
CRISPR/dCas9-mediated LINE-1 inhibition alters immune response gene expression during *Trypanosoma cruzi* infection. (**A**) HEK-293 cells transfected with guide RNAs targeting LINE-1 (sgLINE1) or a non-human genome sequence (sgGAL4) were imaged at 48 hpi. Images were captured at 100× magnification with a 200 ms exposure time. Scale bar, 10 µm. (**B**–**G**) Relative expression of LINE-1 (**B**), MOV-10 (**C**), MDA-5 (**D**), IFN-α (**E**), IFN-β (**F**), and IFN-γ (**G**). Gene expression levels were quantified using the 2^−ΔΔCt^ method and are presented as means ± standard deviation. Statistically significant differences between groups are denoted by different letters. NC: negative control, non-infected cells. +Tc: cells infected with *T. cruzi*. sgLINE-1 + Tc: cells transfected with guide RNAs targeting LINE-1 (sgLINE-1) prior to *T. cruzi* infection.

**Figure 3 ijms-26-04466-f003:**
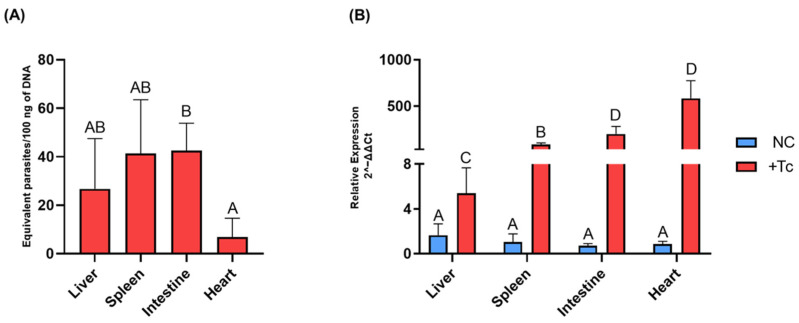
Parasite load and LINE-1 expression in various organs of mice during the chronic phase of Chagas disease. (**A**) Absolute quantification of *T. cruzi* burden in tissue samples, determined by qPCR targeting the Tc40s region of *T. cruzi* cDNA. (**B**) Relative expression of LINE-1 in different organs of BALB/c mice, either infected or uninfected with *T. cruzi*. Gene expression was quantified using the 2^−ΔΔCt^ method. Data are presented as mean ± standard deviation, with different letters indicating statistically significant differences between groups. NC: negative control, non-infected cells. +Tc: cells infected with *T. cruzi*.

**Figure 4 ijms-26-04466-f004:**
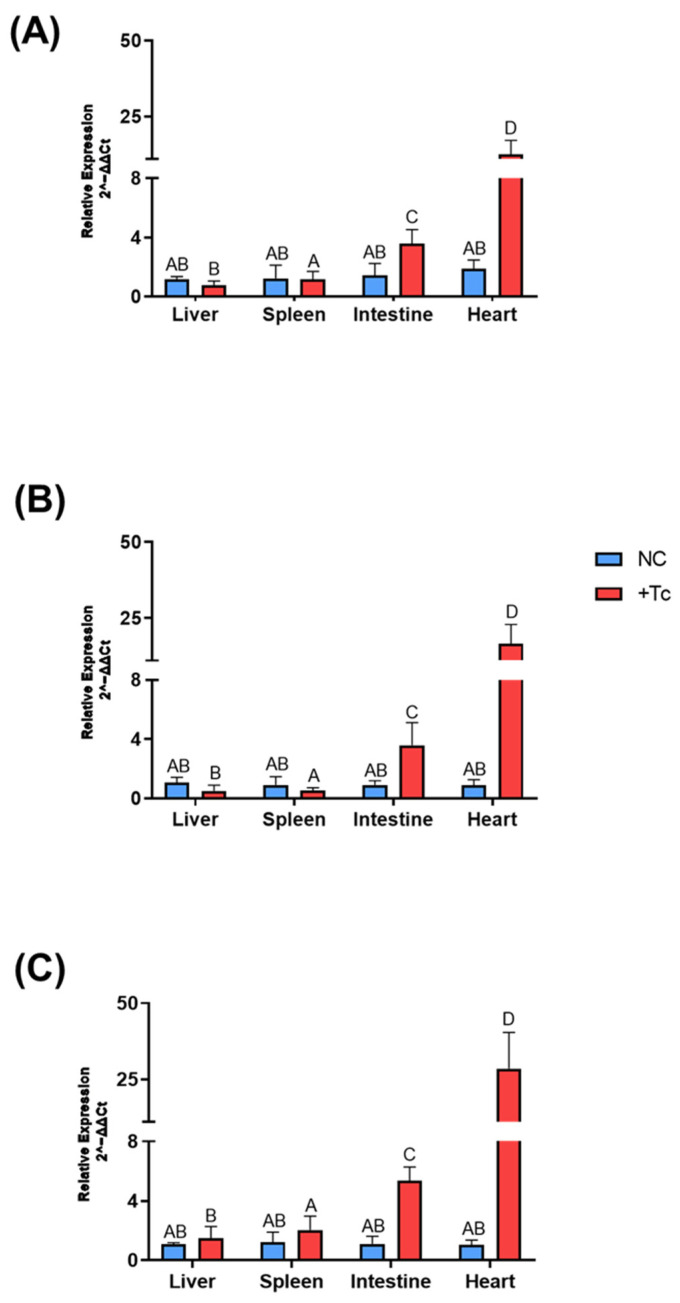
Expression of interferons associated with the immune response in various tissues of BALB/c mice, either infected or uninfected with *Trypanosoma cruzi*. Gene expression levels were quantified using the 2^−ΔΔCt^ method. Data are presented as mean ± standard deviation for both the *T. cruzi*-infected group (+Tc, red bars) and the uninfected control group (NC, blue bars). The analyzed genes include (**A**) IFN-α, (**B**) IFN-β, and (**C**) IFN-γ. Different letters indicate statistically significant differences between samples.

**Figure 5 ijms-26-04466-f005:**
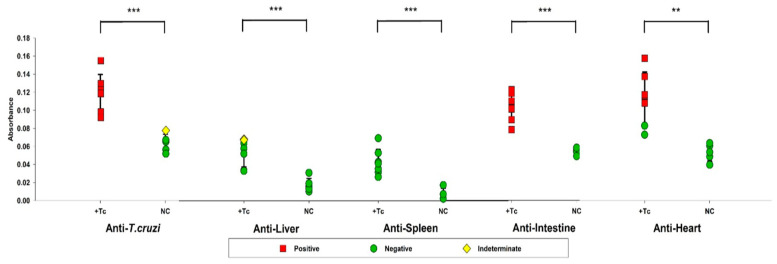
Assessment of antibody production against *Trypanosoma cruzi* antigens and self-antigens from the liver, spleen, intestine, and heart. Serum samples from BALB/c mice, either infected or uninfected with *T. cruzi*, were analyzed by ELISA to quantify IgG antibody levels against *T. cruzi*-specific antigens and organ-derived proteins. Animals with positive serology are represented in red, those with indeterminate serology in yellow, and seronegative animals in green. Data are presented as absolute levels, and bars indicate mean + standard deviation. ** *p* ≤ 0.01, *** *p* ≤ 0.001.

**Figure 6 ijms-26-04466-f006:**
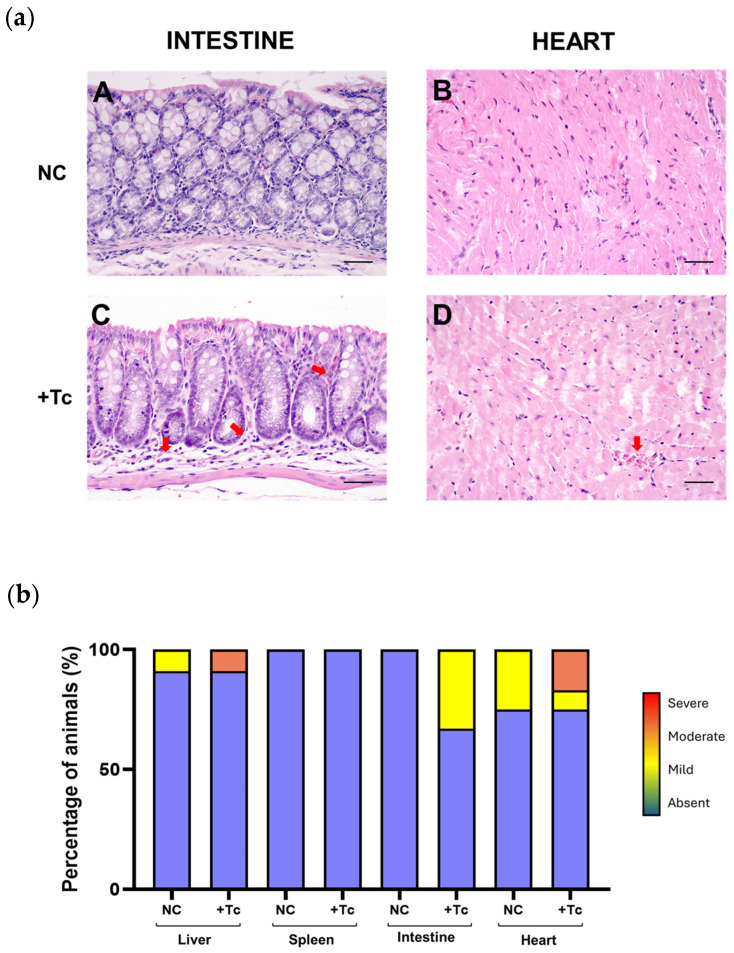
Histopathological assessment of inflammatory infiltrates in *Trypanosoma cruzi*-infected and uninfected animals. (**a**) Representative histological images of intestinal and cardiac tissues from control and infected groups. The upper panel displays images of intestinal (**A**) and cardiac (**B**) tissues from control animals, both exhibiting normal histological architecture without pathological alterations. The lower panel shows intestinal (**C**) and cardiac (**D**) tissues from infected animals, highlighting the presence of inflammatory infiltrates, indicated by red arrows. Hematoxylin and eosin (HE) staining. Scale bar: 100 μm. (**b**) Proportion of animals presenting inflammatory infiltrates in the liver, spleen, heart, and intestine, assessed at 90 dpi in the infected group and at an equivalent time point in the uninfected group. The severity of inflammation was categorized into four grades: absent (blue, 0), mild (yellow, 1), moderate (orange, 2), and severe (red, 3). NC: negative control, non-infected cells. +Tc: cells infected with *T. cruzi*.

**Figure 7 ijms-26-04466-f007:**
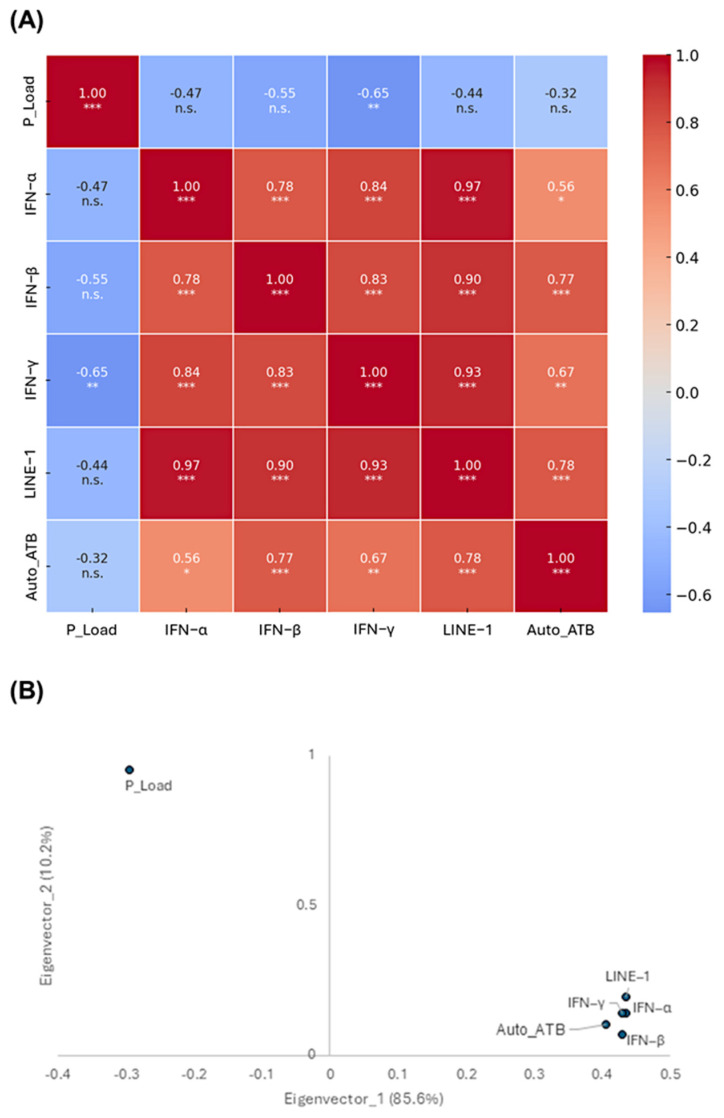
Interaction of LINE-1 expression with parasite- and host-related factors in a murine model of Chagas disease. (**A**) Heatmap illustrates the correlation between parasite load (P_Load), autoantibody titers (Auto_ATB), interferon (IFN) production, and LINE-1 expression. Statistical significance is indicated as follows: * *p* < 0.05, ** *p* < 0.01, *** *p* < 0.001, and non-significant (n.s.). (**B**) Principal component analysis (PCA) eigenvectors representing key parameters associated with Chagas disease pathogenesis. The percentage in parentheses denotes the proportion of variance explained by each eigenvector. Variables positioned within the same quadrant exhibit similar behavior.

## Data Availability

The original contributions presented in the study are included in the article; further inquiries can be directed to the corresponding author.

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
