# Peer review of "Beyond Trypanosoma cruzi: LINE-1 Activation as a Driver of Chronic Inflammation in Chagas Disease"

_ijms, 2025, doi:10.3390/ijms26104466_

Round 1
Reviewer 1 Report
Comments and Suggestions for Authors
Introduction: In the chronic phase of Chagas disease, cardiac symptoms are more frequent than mega-diarrhea syndromes - see line 42.
In the methodology, the authors could explain how trypomastigotes were obtained for the infection of cells and animals.
The meaning of acronyms must be made explicit the first time they are mentioned. For example, DMEM on line 92 and not on line 107.
Figure 4 could be enlarged to be in the same proportion as the other figures.
The authors should explain the choice of the Trypanosoma cruzi strain used in this study. Dr. Sônia Andrade's group has already demonstrated the presence of extensive cardiac lesions in mice infected with the Colombiana strain, which corroborates the greater expression of LINE-1 in the heart.
Author Response
Dear Reviewer,
We sincerely thank you for your valuable comments and constructive suggestions, which have greatly contributed to improving the quality of our manuscript. We remain at your disposal for any further questions or clarifications that may be necessary. Below, we provide our detailed, point-by-point responses and clarifications:
Introduction: In the chronic phase of Chagas disease, cardiac symptoms are more frequent than mega-diarrhea syndromes - see line 42.
The original excerpt was revised to emphasize that cardiac involvement is more prevalent than gastrointestinal manifestations in the chronic phase of Chagas disease:
“During this chronic stage, despite a substantial decrease in parasite burden, patients may develop the so-called “megasyndromes,” which primarily involve the cardiovascular system and, to a lesser extent, the gastrointestinal tract, often resulting in severe organ dysfunction. [2-5].”
In the methodology, the authors could explain how trypomastigotes were obtained for the infection of cells and animals.
To provide a clearer description of the procedures used to obtain trypomastigotes for cell and animal infections, the following excerpt was added to the Methods section:
“Trypomastigote forms were obtained through infection of the L6 cells. Briefly, L6 cells were seeded and cultured in Dulbecco’s Modified Eagle Medium (DMEM) supplemented with 10% (v/v) heat-inactivated fetal bovine serum (FBS), 100 IU/mL penicillin, and 100 µg/mL streptomycin, under a humidified atmosphere of 5% COâ‚‚ at 37°C. Once a confluent monolayer was achieved, the cells were infected with epimastigote forms at a parasite-to-host cell ratio of 5:1. After incubation for ~5 days, trypomastigotes released into the supernatant were harvested, centrifuged at 5,000 rpm for 15 min to remove cell debris, and used for subsequent experimental procedures.”
The meaning of acronyms must be made explicit the first time they are mentioned. For example, DMEM on line 92 and not on line 107.
The adjustment has been made.
Figure 4 could be enlarged to be in the same proportion as the other figures.
Figure 4 has been reformatted to ensure consistency with the proportions of the other figures.
The authors should explain the choice of the Trypanosoma cruzi strain used in this study.
The Colombiana strain of T. cruzi was selected for our study due to its well-characterized myotropic behavior, and this statement has been incorporated into the manuscript.
Dr. Sônia Andrade's group has already demonstrated the presence of extensive cardiac lesions in mice infected with the Colombiana strain, which corroborates the greater expression of LINE-1 in the heart.
We appreciate the insightful comment, which has been addressed and integrated into the revised Discussion section:
Despite that, it is important to highlight that previous studies have reported extensive cardiac lesions in mice infected with the Colombiana strain [72,76]. Furthermore, the inflammatory process was detected specifically in the organs that exhibited a marked increase in L1 expression and IFN production, occurring independently of parasitic load. These findings further support the hypothesis that the L1 retroelement plays a role in modulating the chronic inflammation and autoimmune response in CD.
Reviewer 2 Report
Comments and Suggestions for Authors
Dear Authors, I find your work very interesting; it provides new insights into T. cruzi infections, which, if demonstrated in humans, could lead to alternative therapies for the treatment of CD.
I have some doubts about some of your experiments:
- Experiments with mice can´t be translated to the chronic phase of the disease in humans.
- The number of mice in your experiments is too small to reach any meaningful conclusions, although I understand the possible bioethical concerns.
- The CRISPR inactivation of LINE-1 isn't very clear to me. You have hundreds or thousands of LINE-1 copies. Are you knocking out all ORF2 genes? how many of them? Do you have any DNA-sequencing evidence?
- An interesting assay would be to examine whether injecting dead parasites could trigger the LINE-1 activation
Author Response
Dear Reviewer,
We have addressed the comments raised and incorporated additional information into the manuscript. We remain available to respond to any further questions. We believe the manuscript has been substantially improved, and we are sincerely grateful for your valuable contributions.
Dear Authors, I find your work very interesting; it provides new insights into T. cruzi infections, which, if demonstrated in humans, could lead to alternative therapies for the treatment of CD.
I have some doubts about some of your experiments:
Experiments with mice can´t be translated to the chronic phase of the disease in humans.
We agree with this statement and had already addressed this limitation in our Discussion: 'However, these inflammatory alterations were not as pronounced, which may be attributed to the intrinsic limitations of the murine model in accurately replicating the severity of clinical manifestations observed in human CD [64].' Nevertheless, to make this limitation more explicit, we have added the following sentence to the Conclusion: “In addition, assessing L1 expression in human tissues is crucial to overcome the inherent limitations of murine models in fully recapitulating the pathophysiology of CD.”
The number of mice in your experiments is too small to reach any meaningful conclusions, although I understand the possible bioethical concerns.
In fact, the reduced number of animals used in the study reflects the ethical principles of animal experimentation in our institution. Importantly, this limitation did not compromise the statistical robustness of our analyses. All statistical evaluations yielded satisfactory results, with no indications of data non-convergence or other statistical impediments related to sample size. Analyses performed using SAS software did not generate any warnings regarding model convergence or data integrity.
The CRISPR inactivation of LINE-1 isn't very clear to me. You have hundreds or thousands of LINE-1 copies. Are you knocking out all ORF2 genes? how many of them? Do you have any DNA-sequencing evidence?
The guide RNA was designed to target a region within the ORF2 of LINE-1 that encodes the reverse transcriptase domain, which is known to be conserved among functional LINE-1 copies (Clements & Singer, 1998). Therefore, although sequence variations and recombination events may occur in some genomic LINE-1 copies, our CRISPR/dCas9 system is expected to act on a substantial number of elements. It is not feasible to determine the exact number of copies inactivated by the CRISPR/dCas9 system; thus, quantitative PCR was employed to assess the overall repression of LINE-1 expression. To clarify this point, we have included the following sentences in the manuscript:
“Double-stranded single guide RNAs (sgRNAs) were designed to specifically target the conserved region encoding the reverse transcriptase domain within ORF-2 region of L1 sequences.”
“Quantitative PCR (qPCR) analyses were performed to verify the efficacy of LINE-1 inactivation mediated by the CRISPR/dCas9 system.”
An interesting assay would be to examine whether injecting dead parasites could trigger the LINE-1 activation
We thank the reviewer for this insightful suggestion, which highlights a highly relevant experimental approach to further explore L1 activation in the context of host–parasite interactions in Chagas disease. This point has been incorporated into the manuscript as a perspective: “Future studies should investigate whether L1 activation can also be triggered by different strains, non-viable parasites or by parasite-derived extracellular vesicles.”
Reviewer 3 Report
Comments and Suggestions for Authors
The manuscript by Dias et al. investigated the role of the LINE-1 transposon during the infection of T. cruzi using in vitro and in vivo models. Their analyses included analyzing and correlating other genes associated with the transposon and the immune response (such as IFN signaling pathways).
The results show how L1 activation may influence host-pathogen interactions and immune dysregulation during infection and disease progression.
Overall, the manuscript is well written and shows strong evidence, and somehow, their discussions and conclusions are well supported by their results.
I only have a few minor suggestions to improve this manuscript:
Many references are missing or too old, especially in the introduction and discussion sections. All sentences where relevant mechanisms, results and previous evidence have been provided must be cited and/or updated (e.g., lines 53-55, 61-63, 281-283, etc.).
Please revise the entire manuscript and include any missing references accordingly.
Remove "arbitrary units" in all figures showing qPCR results. Or please specify the criteria for this assumption.
Without new evidence (using up-to-date techniques) supporting the hypotheses raised more than 10 years ago about the possibility of kDNA sequences being transferred to the host genome, I suggest removing that statement (lines 291-293). Besides, this manuscript did not provide evidence supporting this mechanism.
Please provide more information about the chronic stage in the materials and methods section. Why is 90 considered enough to reach the chronic stage? Do the authors use any markers to confirm acute/chronic stages? Do they identify any evident tissue/skin damage/weight/anomalies (organomegaly) after 90 days? Did all of the infected animals survive the infection?
Author Response
Dear Reviewer,
We sincerely thank you for your valuable comments and constructive suggestions, which have greatly contributed to improving the quality of our manuscript. We remain at your disposal for any further questions or clarifications that may be necessary. Below, we provide our detailed, point-by-point responses and clarifications:
The manuscript by Dias et al. investigated the role of the LINE-1 transposon during the infection of T. cruzi using in vitro and in vivo models. Their analyses included analyzing and correlating other genes associated with the transposon and the immune response (such as IFN signaling pathways).
The results show how L1 activation may influence host-pathogen interactions and immune dysregulation during infection and disease progression.
Overall, the manuscript is well written and shows strong evidence, and somehow, their discussions and conclusions are well supported by their results.
I only have a few minor suggestions to improve this manuscript:
Many references are missing or too old, especially in the introduction and discussion sections. All sentences where relevant mechanisms, results and previous evidence have been provided must be cited and/or updated (e.g., lines 53-55, 61-63, 281-283, etc.).
Please revise the entire manuscript and include any missing references accordingly.
The manuscript has been thoroughly revised, and additional references have been incorporated to support and strengthen the revised content.
Remove "arbitrary units" in all figures showing qPCR results. Or please specify the criteria for this assumption.
The adjustments have been made.
Without new evidence (using up-to-date techniques) supporting the hypotheses raised more than 10 years ago about the possibility of kDNA sequences being transferred to the host genome, I suggest removing that statement (lines 291-293). Besides, this manuscript did not provide evidence supporting this mechanism.
We appreciate the reviewer’s thoughtful comment. We would like to clarify that the manuscript presents this possibility merely as a hypothetical alternative. We believe that maintaining this discussion could stimulate further investigations by other research groups equipped with advanced techniques to explore this intriguing topic. Nonetheless, to ensure that it is clear that kDNA integration is discussed only as a hypothetical scenario, we have revised the text accordingly:
“Additionally, a hypothetical alternative mechanism could involve the transfer of mitochondrial DNA (kDNA) sequences from the parasite to the host genome, considering that L1 elements have been identified as potential major integration sites for kDNA [42,43].”
Please provide more information about the chronic stage in the materials and methods section. Why is 90 considered enough to reach the chronic stage? Do the authors use any markers to confirm acute/chronic stages? Do they identify any evident tissue/skin damage/weight/anomalies (organomegaly) after 90 days? Did all of the infected animals survive the infection?
The characterization of 90 dpi as an early chronic phase of Chagas disease in murine models is well established in the literature (Hassan et al., 2006; Vilar-Pereira et al., 2016; Arias-Argáez et al., 2023; Rios et al., 2023). At this stage, a significant reduction in parasite burden is typically observed, accompanied by the production of anti-T. cruzi IgG antibodies—markers of chronic infection that were also used in our study. In addition, some animals may begin to exhibit tissue alterations consistent with early chronic pathology. To further clarify this point, we have included the following sentence in the Methods section: “For in vivo studies, sample collection occurred at the early chronic phase (Arias-Argáez et al., 2023; Rios et al., 2023), at 90 dpi, or at a corresponding time for uninfected animals.”
In our experimental conditions, none of the animals exhibited megasyndromes, and more evident tissue damage was not detected . This finding is addressed in the Discussion: “However, these inflammatory alterations were not as pronounced, which may be attributed to the intrinsic limitations of the murine model in accurately replicating the severity of clinical manifestations observed in human CD [64]. Additionally, the animals were still in the early stages of the chronic phase, a period in which overt clinical manifestations are not yet fully established [65,66].” Furthermore, no mortality was observed among infected animals during the experimental period.
Arias-Argáez BC, Dzul-Huchim VM, Haro-Álvarez AP, Rosado-Vallado ME, Villanueva-Lizama L, Cruz-Chan JV, Dumonteil E. Signature of cardiac alterations in early and late chronic infections with Trypanosoma cruzi in mice. PLoS One. 2023 Oct 5;18(10):e0292520. doi: 10.1371/journal.pone.0292520. PMID: 37797045; PMCID: PMC10553825.
Hassan GS, Mukherjee S, Nagajyothi F, Weiss LM, Petkova SB, de Almeida CJ, Huang H, Desruisseaux MS, Bouzahzah B, Pestell RG, Albanese C, Christ GJ, Lisanti MP, Tanowitz HB. Trypanosoma cruzi infection induces proliferation of vascular smooth muscle cells. Infect Immun. 2006 Jan;74(1):152-9. doi: 10.1128/IAI.74.1.152-159.2006. PMID: 16368968; PMCID: PMC1346667.
Rios LE, Lokugamage N, Garg NJ. Effects of Acute and Chronic Trypanosoma cruzi Infection on Pregnancy Outcomes in Mice: Parasite Transmission, Mortality, Delayed Growth, and Organ Damage in Pups. Am J Pathol. 2023 Mar;193(3):313-331. doi: 10.1016/j.ajpath.2022.11.010. Epub 2022 Dec 21. PMID: 36565805; PMCID: PMC10013038.
Vilar-Pereira G, Carneiro VC, Mata-Santos H, Vicentino AR, Ramos IP, Giarola NL, Feijó DF, Meyer-Fernandes JR, Paula-Neto HA, Medei E, Bozza MT, Lannes-Vieira J, Paiva CN. Resveratrol Reverses Functional Chagas Heart Disease in Mice. PLoS Pathog. 2016 Oct 27;12(10):e1005947. doi: 10.1371/journal.ppat.1005947. PMID: 27788262; PMCID: PMC5082855.